# Random Forests for Landslide Prediction in Tsengwen River Watershed, Central Taiwan

**Youg-Sin Cheng** [1,2] 🆔, **Teng-To Yu** [1,*] **and Nguyen-Thanh Son** [2] 🆔

1 Department of Resources Engineering, National Cheng Kung University, Tainan City 701, Taiwan; yscheng@csrsr.ncu.edu.tw

2 Center for Space and Remote Sensing Research, National Central University, Taoyuan City 320, Taiwan; ntson@csrsr.ncu.edu.tw

* Correspondence: yutt@mail.ncku.edu.tw

**Abstract:** Landslides have been identified as one of the costliest and deadliest natural disasters, causing tremendous damage to humans and societies. Information regarding the spatial extent of landslides is thus important to allow officials to devise successful strategies to mitigate landslide hazards. This study aims to develop a machine-learning approach for predicting landslide areas in the Tsengwen River Watershed (TRW), which is one of the most landslide-prone areas in Central Taiwan. Various spatial datasets were collected from 2009 to 2015 to derive 36 predictive variables used for landslide modeling with random forests (RF). The results of landslide prediction, compared with ground reference data, indicated an overall accuracy of 91.4% and Kappa coefficient of 0.83, respectively. The findings achieved from estimates of predictor importance also indicated to officials that the land-use/land-cover (LULC) type, distance to previous landslides, distance to roads, bank erosion, annual groundwater recharge, geological line density, aspect, and slope are the most influential factors that trigger landslides in the study region.

**Keywords:** landslide prediction; random forests (RF); LiDAR; Tsengwen River Watershed

## 1. Introduction

Landslides are the fifth-deadliest natural disaster after windstorms, floods, earthquakes, and extreme temperature during the last 20 years [1]. Globally, they occur over a wide range of spatial and temporal scales across mountainous landscapes. Landslides are driven by gravity and are characterized by movements of solid rock, debris, and soil, causing severe human casualties, property loss, and infrastructure and environmental damage [2–4]. Landslides are often triggered by other natural disasters, including heavy precipitation and earthquakes. In recent years, due to the impacts of climate change and rapid population growth, coupled with the increased frequency and intensity of typhoons, storms, and hurricanes, landslide-related casualties have significantly increased, especially in less-developed and developing countries where slope cultivation and agriculture are commonly practiced [5–7].

Landslides are also severe in Taiwan and considered a major natural hazard in hilly and mountainous regions. They cause risk to life and infrastructure and are difficult to predict. Taiwan's mountainous area accounts for roughly 70% of the country's total land area. The majority of the population is concentrated in alluvial plains and basins. Due to population growth, pressures on narrow residential areas often lead to urban expansion into hilly and sloped areas that are critically prone to landslides. On the other hand, because Taiwan is located at the subduction zone of the Philippine Sea and Eurasian Plates, collisions between these two plates have resulted in many folds and faults in the formed mountains. In addition, the country often experiences typhoons and storms during the rainy season due to the effects of the western North Pacific typhoon belt. Along with a lack of ecological protection and soil conservation as well as improper hillside development for

agriculture, landslides often occur, blocking roadways, isolating mountain areas, injuring people, and damaging property [8].

A number of studies using machine-learning algorithms, such as random forests (RF), support vector machines (SVMs), and decision trees, have been implemented for landslide monitoring [9–11]. The RF method, recently developed to deal with large, complex data, is considered to be a superior classification algorithm and has been widely used in various fields of data mining [12,13]. Various parameters are often adopted to analyze landslide susceptibility, such as geology, soil depth, soil type, and land-use/land-cover (LULC) types [14–17]. However, the use of these datasets usually reveals the limitations of availability and scale [18]. Therefore, the evaluation of landslide areas based on a digital elevation model (DEM) has been widely conducted, assuming that topographical factors are more efficient. The LiDAR data are thus used in this study to generate high-resolution DEM data. Studies also found that landslide prediction performance decreases with finer resolutions [19], while changes in landside prediction parameters with a spatial scale have a significant effect on the results [20–22].

The main objective of this study is to develop a RF-based approach using different spatial datasets with different scales to predict landslide areas in the Tsengwen River Watershed (TRW), Taiwan. The results in the form of spatial and quantitative landslide information could be valuable for devising successful institutional interventions and management plans to minimize the risk of landslides in the region.

## 2. Study Area

The study region, located upstream of TRW, Taiwan, covers approximately 40 km$^2$ (Figure 1). The elevation of the region ranges between 700 and 2400 m above sea level. More than 20% of the total area has a slope greater than 45°. The region is surrounded by the Naipan, Shuisheliao, and Kantzuchai faults, the lithology of which is characterized by shale, sandstone, and interlayered sandstone and mudstone from the mid-Miocene to Pleistocene [23]. The average annual precipitation at the Leye meteorological station (low altitude) and Alishan station (high altitude) is approximately 3382 mm and 4267 mm, respectively. Landslides, recognized as one of the costliest natural disasters in the region, frequently occur in response to a number of natural events, including heavy rainfall and earthquakes, and anthropogenic events, such as building or road construction. For example, the area of landslides in the study region, caused by typhoon Morakot on 8 August 2009, was roughly 1.2 km$^2$ during 2009–2010, while the new landslide areas in 2015 increased at least 0.04 km$^2$.

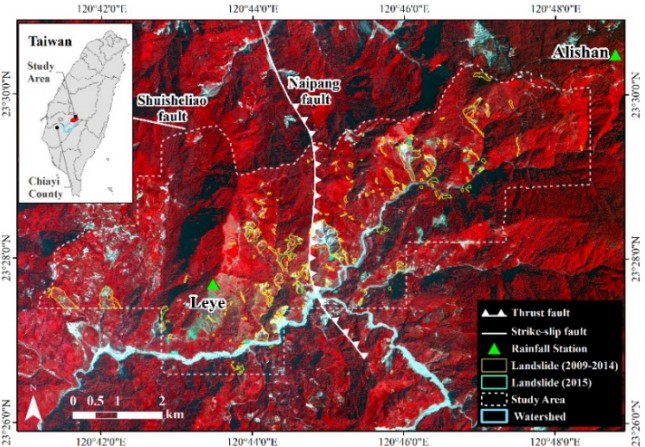

**Figure 1.** Map of the study region showing the reference landslide areas during 2009−2015 used for RF model training and accuracy assessment with reference to the false-color SPOT-5 image on 5 March 2015 (RGB = 3, 2, 1).

## 3. Data Collection

A suit of datasets, collected from the Council of Agriculture and the Central Geological Survey, Taiwan, was used for landslide prediction, including landslide inventory maps, LULC maps, geological maps, LiDAR data (March 2015), SPOT5 (March 2015), and precipitation data. The landslide maps collected from the Forestry Bureau and Council of Agriculture, Taiwan, were primarily constructed using high resolution satellite images from 2009 to 2015 with an area greater than 0.1 ha. The landslide patches had been validated using field survey data conducted by the Central Geological Survey, Taiwan. Lastools software was used to generate the digital elevation model (DEM) (1 m resolution) for topographic analysis. The precipitation data from 14 weather stations around the study region, collected from the Taiwan Central Weather Bureau during 2011 to 2014, were used to evaluate the effects of annual rainfall on groundwater charging.

## 4. Methodology

A flowchart of the methodology illustrates the main steps in data processing used in this study to predict landslide areas in the region (Figure 2).

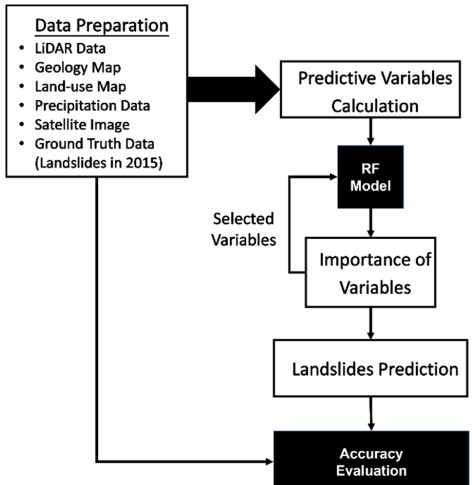

**Figure 2.** Flowchart of the methodology used for landslide prediction in the study region.

### 4.1. Random Forests

The RF algorithm [24], which is an ensemble learning method developed by combining predictions from decision trees, was applied to build the regression or classification model. This method has several advantages. For example, it is a more commonly used in a multivariate regression or classification method, and assumptions on the distribution of explanatory variables are not required. Likewise, the mixed use of categorical and numerical variables without resorting to the use of indicator (or dummy) variables is allowed, and interactions and non-linearities between variables can be accounted for [25]. The RF has been widely applied in many fields with excellent performance [13,26,27]. It has also achieved good results in landslide prediction and landslide location identification [26,28]. In this study, the KNIME Analytics Platform was used to construct the RF prediction model. The data were divided into two groups of pixels, namely training pixels (30%) and evaluation pixels (70%), for accuracy assessment. For building the RF model, 50% of the group of training pixels were randomly extracted and used as training samples for model establishment, leaving 50% of the data for model validation.

### 4.2. Predictive Variables

Predictor variables used in this study can be categorized into three groups, namely topographic, hydrological, and geological (Table 1). Topographic variables, derived from DEM, include aspect, river density, slope, curvature, wetness, and roughness. Hydrological

variables, derived from GIS layers of river and road networks, include the distance to banks of rivers, distance to roads, and river density. Geological variables, derived from geological maps, comprise geological density line, distance to banks of old landsides, and distance to dip slope. The slope, curvature, wetness, and roughness were calculated for different moving window sizes (i.e., $3 \times 3$, $5 \times 5$, $7 \times 7$, $9 \times 9$, and $11 \times 11$) to investigate the scale effects of these topographic variables on the results. Likewise, different radiuses (R) for geological line density and river density were also calculated for the analysis of search effects using the Line Density tool in ArcGIS. The LULC classes, Normalized Difference Vegetation Index (NDVI), and annual ground water recharge were used. In total, 36 variables were derived and used in this study for building a landslide prediction model. Figure 3 shows the spatial variations of predictor variables in this study. Specifically, the slope and curvature were calculated using the quadratic approximation method [29], and the wetness factor quantifying the topographic control of the hydrological processes was estimated from the catchment area (*As*) using terrain analysis using DEM as follows:

$$Wetness \ = \ ln(\frac{As}{tan(slope)}) \tag{1}$$

**Table 1.** Predictive variables derived from various data sources used in this study for landslide prediction.

| Data Source | Data Type | Variables |
|---|---|---|
| LiDAR data (March 2015) | DEM (1 m resolution) | Aspect |
| | | Slope (moving window sizes: $1 \times 1$, $3 \times 3$, $5 \times 5$, $7 \times 7$, $9 \times 9$, and $11 \times 11$) |
| | | Curvature (moving window sizes: $1 \times 1$, $3 \times 3$, $5 \times 5$, $7 \times 7$, $9 \times 9$, and $11 \times 11$) |
| | | Roughness (moving window sizes: $3 \times 3$, $5 \times 5$, $7 \times 7$, $9 \times 9$, and $11 \times 11$) |
| | | Wetness (moving window sizes: $1 \times 1$, $3 \times 3$, $5 \times 5$, $7 \times 7$, $9 \times 9$, and $11 \times 11$) |
| | | River Density (radii: R = 50, R = 100, and R = 500) |
| Satellite image | SPOT5 (5 March 2015) | NDVI |
| Precipitation data | Distribution of annual precipitation data from 14 weather stations (2011–2014) | Average annual groundwater recharge |
| Geology map | Faults and folds | Geological line density (radii: R = 100 and R = 500) |
| | Dip slope | Distance to dip slope |
| Land-use map | LULC types | LULC types (12 classes) |
| | Location of roads | Distance to roads |
| | Location of rivers | Distance to river banks |
| Landslide inventory map | Old landslide areas (before March 2015) | Distance to old landslides |
| | Landslide areas in 2015 | Landslides in 2015 |

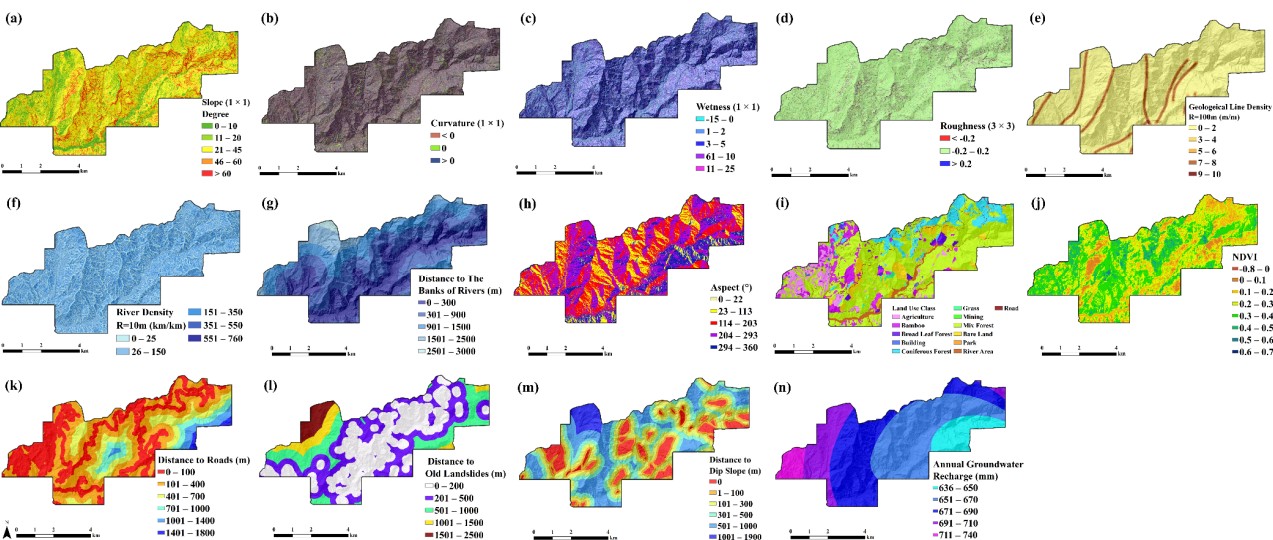

**Figure 3.** Variables used for landslide modeling in this study: (**a**) slope, (**b**) curvature, (**c**) wetness, (**d**) roughness, (**e**) geological line density, (**f**) river density, (**g**) distance to the banks of rivers, (**h**) distribution of slope aspect, (**i**) LULC types, (**j**) distribution of NDVI, (**k**) distance to roads, (**l**) distance to old landslides, (**m**) distance to the dip slope area, and (**n**) groundwater recharge.

The roughness and density of the geological line structures, respectively, characterizing the surface characteristics of the terrain, were calculated from DEM [30]. The river density, indicating the drainage system, was also calculated from DEM using ArcGIS's hydrology toolset.

The distance to river banks, distance to roads, and distance to old landslides, characterizing the erosion extension and the strength and stability at the foot of the slope, were calculated using the Euclidean distance. The aspect, calculated from DEM using ArcGIS's aspect tool, indicates a slope's horizontal direction [31], potentially causing landslides with increasing probability by torrential rain or incident solar radiation [32,33]. The LULC types (12 classes) were extracted from the LULC map, and NDVI was calculated from the SPOT-5 image. The distance to the dip slope was adopted as one variable by the dip slope map from the Taiwan Ministry of Economic Affairs [23]. The groundwater level of the slope can be estimated from long-term annual precipitation data, evapotranspiration data, and base flow data [34] using the following equation:

$$QR = BFI \times (P - ET) \tag{2}$$

where QR, BFI, P, and ET are groundwater recharge, base-flow index, precipitation, and evapotranspiration, respectively.

### 4.3. Accuracy Assessment

The Gini index algorithm [35] was used for evaluating the importance of 36 predictor variables. The variables that had an importance level greater than 0.1 were used to build the RF model. The results of landslide prediction achieved from this model were also compared with those from the model using 36 variables for the sake of performance comparison between these two models. For accuracy assessment of the predicted landslide areas, the receiver operation curve (ROC) was applied to evaluate the prediction ability of the model. The area under the ROC (AUC) with the higher value indicates a more effective prediction [36]. A pixel-by-pixel comparison between the predicted landslide areas and the ground reference data was also implemented using a group of evaluation pixels (i.e., 70% of the reference landslide map). To facilitate this comparison, the predicted landslide results were first categorized into the two classes of landside areas and non-landslide areas. The results of the landslide prediction were then verified by comparing pixels from the group of pixels (used for accuracy assessment) with those synchronized from the RF prediction map. The error matrix, using overall producer and user accuracies, and the Kappa coefficient were calculated to measure the prediction accuracy.

## 5. Results and Discussion

### 5.1. Importance of Predictive Variables

The results of the sensitivity analysis indicated that 11 out of the 36 variables had probability values greater than 0.1 (Table 2). They were the LULC type, recharge of ground water, distance to river banks, distance to old landslides, distance to dip slope, geological line density (R = 100 m), distance to roads, geological line density (R = 500 m), river density (R = 100 m), aspect, and slope (3 × 3). The LULC type, recharge of ground water, and distance to river banks were the three most important variables for landslide modeling, with probability values ranging from 0.47 to 0.51. The reasons were that urbanization accelerated the change of LULC type in mountainous areas, transforming soil erosion resistance ability and permeability simultaneously [37], while the recharge of ground water and the shorter distance to river banks [38] could respectively change the pore pressure and increase soil erosion in slope areas, consequently triggering landslides [39,40]. The distances to old landslides and dip slope had the same value of 0.3, which was attributable to the fact that the areas surrounded by old landslides and dip-slope landslides were highly concentrated [41] due to the strike of the slope face and the strata being parallel to the intersection angle within 20° [42], which might potentially cause dip-slope landslides.

**Table 2.** Results of sensitivity analysis for evaluating variables used in RF modeling of landslides.

| Rank | Variables | Importance | Rank | Variables | Importance |
|------|-----------|------------|------|-----------|------------|
| 1 | LULC types | 0.51 | 19 | Roughness (9 × 9) | 0.06 |
| 2 | Recharge of ground water | 0.50 | 20 | Slope (9 × 9) | 0.06 |
| 3 | Distance to the bank of rivers | 0.47 | 21 | Wetness (3 × 3) | 0.06 |
| 4 | Distance to old landslides | 0.32 | 22 | Wetness (9 × 9) | 0.05 |
| 5 | Distance to dip slope | 0.32 | 23 | Slope (5 × 5) | 0.05 |
| 6 | Geological line density (R = 100 m) | 0.22 | 24 | Wetness (11 × 11) | 0.05 |
| 7 | Distance to roads | 0.22 | 25 | Curvature (9 × 9) | 0.04 |
| 8 | River density (R = 100 m) | 0.22 | 26 | Roughness (11 × 11) | 0.04 |
| 9 | Geological line density (R = 500 m) | 0.18 | 27 | Roughness (5 × 5) | 0.04 |
| 10 | Aspect | 0.14 | 28 | Curvature (1 × 1) | 0.04 |
| 11 | Slope (3 × 3) | 0.14 | 29 | Roughness (7 × 7) | 0.02 |
| 12 | Slope (11 × 11) | 0.08 | 30 | Roughness (3 × 3) | 0.02 |
| 13 | Slope (7 × 7) | 0.08 | 31 | Curvature (3 × 3) | 0.01 |
| 14 | NDVI | 0.08 | 32 | River Density (R = 10 m) | 0.01 |
| 15 | Slope (1 × 1) | 0.08 | 33 | Curvature (5 × 5) | 0.01 |
| 16 | River density (R = 50 m) | 0.08 | 34 | Curvature (11 × 11) | 0.01 |
| 17 | Wetness (7 × 7) | 0.08 | 35 | Wetness (1 × 1) | 0.01 |
| 18 | Wetness (5 × 5) | 0.07 | 36 | Curvature (7 × 7) | 0.00 |

The probability values of 0.2 were observed for geological line density (R = 100 m), distance to roads, river density (R = 100 m), and geological line density variables. Geological factors related to faults and folds are highly sensitive for predicting landslides. Likewise, the development of roads in mountainous areas, where the density of debris flows was especially observed, could decrease the stability at the foot of the slope and influence flow paths [43], thereby increasing the probability of landslides. The topographic factors of aspect and slope (3 × 3) revealed the lower value of 0.14, because the horizontal direction was windward or to the sunny side [31] in the study area, and a steep slope with a windward or sunny side aspect could cause potential landslides with increasing probability caused by torrential rain or incident solar radiation [32,33]. Broadly speaking, NDVI and topographical variables (wetness, roughness, and curvature) were less important than LULC types, or hydrological and geological factors for RF modeling of landslides. Moreover, the results also revealed that a coarser window size, such as slope (3 × 3), wetness (7 × 7), and roughness (9 × 9) or longer search radiuses, such as river density (R = 100 m), could yield more accurate predictions of landside areas. The topographical variables (1 m resolution) were not identified as effective and sensitive variables for landslide modeling, although they spatially characterized more details of the study region. The larger moving windows, which could filter the noise of terrain data, were deemed more effective for modeling of potential landslide areas.

*5.2. Accuracies of RF Model and Predicted Landslide Results*

The results, achieved from the RF model validation using 36 and 11 variables indicated close agreement between the predicted and actual values in both cases (Table 3). The overall accuracies and Kappa coefficients obtained for the model using 36 variables were 99.5% and 0.99, while slightly more accurate results were observed for the model using 11 values with an overall accuracy of 99.7% and 0.99, respectively. The use of 11 variables for RF modeling could improve the landslide prediction of 0.2% and 0.1% of the overall and user's accuracies, respectively. The model coefficients, obtained from the model using 11 variables derived from the sensitivity analysis, were used for predicting susceptibility landslide areas. The predicted results with probabilistic values from 0.5 to 1 could be categorized into three classes to depict landslide susceptibility in the study region, including low susceptibility (0.5–0.7), moderate susceptibility (>0.7–0.9), and high susceptibility (>0.9–1) (Figure 4). In general, the distributions of predicted susceptibility landslide areas were spatially comparable with those from the ground reference map (Figure 1). The high

susceptibility landslide areas were mostly concentrated in relatively steep slopes located along main roads (Figure 4), where sites composed of weak or fractured materials could easily experience landslides due to heavy rain. An example illustrates three enlarged sites (A, B, and C) and a high density landside region (green rectangle), which were located in steep slopes along main roads (Figures 4 and 5). Broadly speaking, the high susceptibility landslide areas were accurately predicted from the RF model using 11 selected variables showing spatially comparable distributions with the ground reference data.

**Table 3.** Results of RF model accuracy evaluation and error verification of the predicted landslide map.

| Accuracy | Landslide | | Non-Landslide | | Overall Accuracy (%) | Kappa Coefficient |
|---|---|---|---|---|---|---|
| | User's Accuracy (%) | Producer's Accuracy (%) | User's Accuracy (%) | Producer's Accuracy (%) | | |
| RF model (36 variables) | 99.2 | 99.8 | 99.8 | 99.2 | 99.5 | 0.99 |
| RF model (11 variables) | 99.4 | 99.9 | 99.9 | 99.4 | 99.7 | 0.99 |
| Error verification | 99.2 | 83.5 | 85.7 | 99.3 | 91.4 | 0.83 |

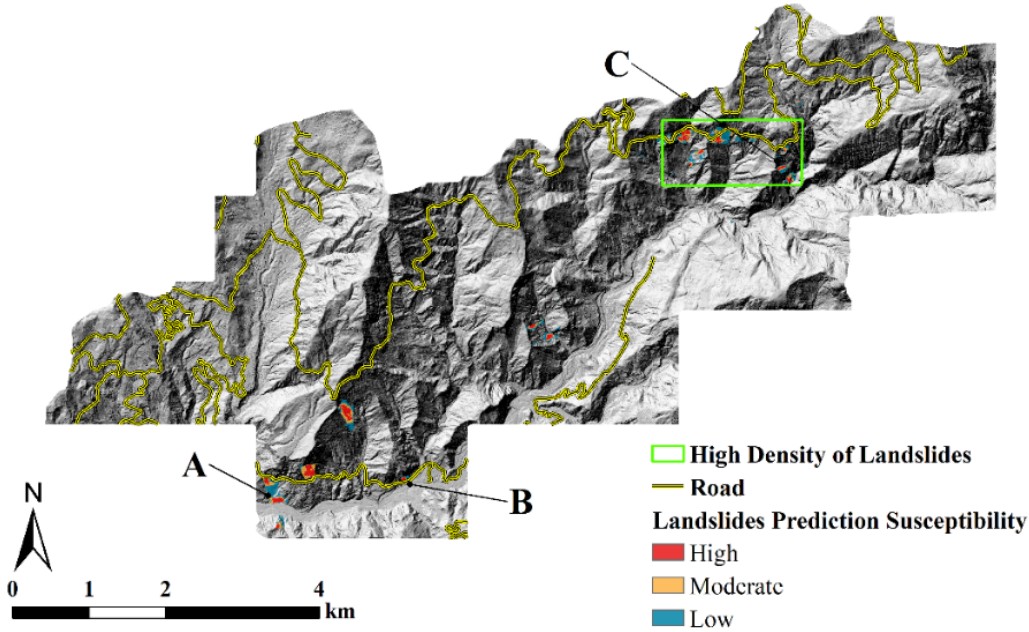

**Figure 4.** Spatial distributions of predicted susceptibility landslide areas in the study region. The enlarged sites and a green rectangle region were chosen to illustrate high density landslides in steep slope areas.

The predicted results of high-susceptibility landslides, compared with the ground reference data (i.e., group of pixels used for accuracy assessment), indicated the satisfactory overall accuracy of 91.4% and the AUC of 98%. The Kappa coefficient, measuring the difference between actual agreement and agreement expected by chance, was 0.8, confirming the reliability of the RF model for landslide modeling. Of the 70% of the reference pixels used to measure the per-class accuracy of predicted landslide and non-landslide areas, the producer's accuracy was 99.2%, indicating that only 0.8% of the landslide pixels were omitted from the interpretation results. The lower user accuracy was observed for the non-slide class (85.7%), a corollary to the commission error of 14.3%. For instance, the three enlarged sites (A, B, and C) showed misclassified landslide areas (commission errors) between the predicted landslides and the ground reference landslides in 2015 (Figure 5). The reasons for such commission errors were mainly attributed to the resolution bias between the ground reference map and the predicted landslide map. In this study, the ground reference data were constructed from high resolution aerial photos and field surveys, while the predicted landslide map was generated by the RF model using predictor variables

spatially resampled from various datasets with the same or coarser spatial resolutions. The accuracy assessment performed using data with different spatial resolutions could exaggerate the commission and omission errors of the predicted landslide results.

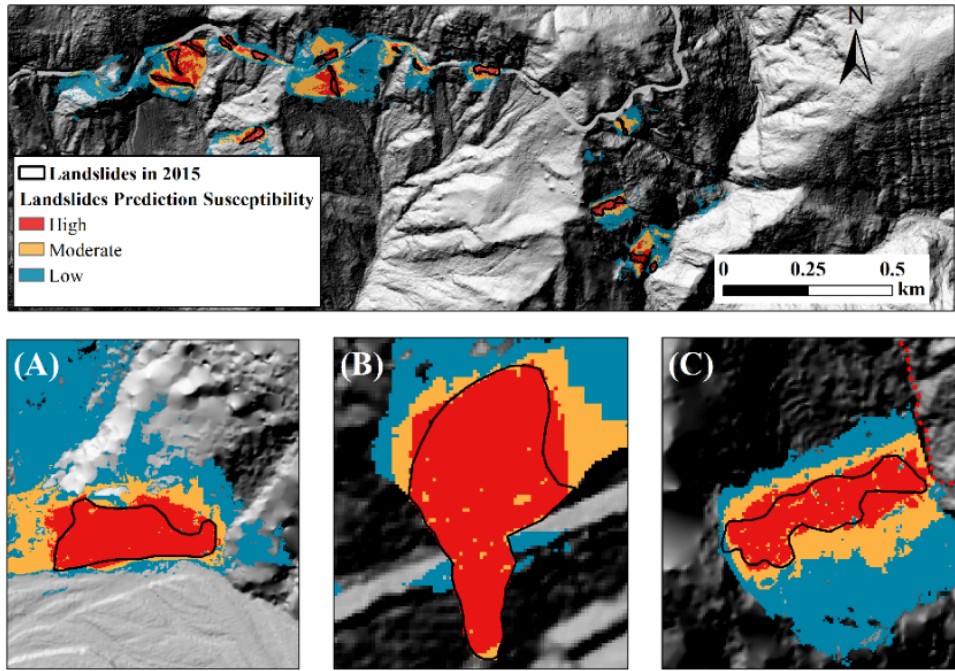

**Figure 5.** A high-density landside region and three enlarged sites A, B, and C of predicted landsides along the Alishan highway, overlaid with the ground reference landslides in 2015.

## 6. Conclusions

The findings achieved from this study validated our RF-based approach for landslide prediction in TRW, Taiwan, using optimal hydrological, geological, and topographical variables and moving window sizes. The results of the sensitivity analysis indicated that 11 out 36 variables, including LULC types, recharge of ground water, distance to the banks of rivers, distance to old landslides, distance to dip slope, geological line density (R = 100 m), distance to roads, geological line density (R = 500 m), river density (R = 100 m), aspect, and slope (3 × 3), were more important factors in the landslide prediction model. The results of landslide prediction, verified with the ground reference data, indicated an AUC of 98%, overall accuracy of 91.4%, and a Kappa coefficient of 0.83, respectively. Although the resolution bias between the ground reference and predicted maps exaggerated the landslide mapping accuracy, the methods used in this research work could provide reliable spatial and quantitative information on landslides in the study region. Such information could be valuable to policymakers to evaluate and formulate their interventions for landslide preparedness, mitigation, and management.

**Author Contributions:** Conceptualization: Y.-S.C. and T.-T.Y.; methodology: Y.-S.C.; validation: Y.-S.C. and N.-T.S.; data curation: Y.-S.C.; result analysis: Y.-S.C.; writing—original draft preparation: Y.-S.C.; writing—review and editing: N.-T.S. and T.-T.Y.; visualization: Y.-S.C.; supervision: T.-T.Y. with contribution from all coauthors. All authors have read and agreed to the published version of the manuscript.

**Funding:** This research received no external funding.

**Institutional Review Board Statement:** Not applicable.

**Informed Consent Statement:** Not applicable.

**Data Availability Statement:** The data are not publicly available due to copyright.

**Acknowledgments:** The LiDAR datasets were provided by the Chung Hsing Surveying Company Limited, respectively.

**Conflicts of Interest:** The authors declare no conflict of interest.

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
