# Peer review of "Random Forests for Landslide Prediction in Tsengwen River Watershed, Central Taiwan"

_remotesensing, doi:10.3390/rs13020199_

Round 1

Reviewer 1 Report

The paper develops a model predicting landslide susceptibility using random forests in a specified region. Thus the paper is an application paper with little innovation. Tables and figures of the paper are adequate. However, the paper is generally well-written and an interesting one. Regarding the notation used, the 3X3, 9X9 etc symbols used must be better described and specify their dimensions, if any. More importantly, regarding the variables in the analyses it is desirable to compare them with those used in previous similar analyses (Taalab et al, 2018, Xiong et al. 2020). Regarding the obtained results, it is also is desirable to compare them with those of previous similar analyses. Furthermore it is important to discuss how the critical variables obtained relate to the underlying physical processes. In this it can be stated that landslides stability is affected by high slope, rainfall, loose soil, high water table line etc. (Bordonia et al, 2015, Heshmati et al., 2011). Finally, as the study covers only 6 years, where strong earthquakes may not have occurred, the authors should discuss if it is expected that the critical variables may change if a different time period is considered where a strong earthquake occurs.

REFERENCES

Bordonia M., Meisina, R.Valentino ,N.Lu, M. Bittelli, S. Chersich. Hydrological factors affecting rainfall-induced shallow landslides: From the field monitoring to a simplified slope stability analysis. Engineering Geology, 2015, 193, 19-37

Heshmati M., A.Arifin, J.Shamshuddin, N.M.Majid, M.Ghaituri, Factors affecting landslides occurrence in agro-ecological zones in the Merek catchment, Iran. Journal of Arid Environments, Volume 75, Issue 11, November 2011, Pages 1072-1082

Taalab K., T. Cheng, Y. Zhang. Mapping landslide susceptibility and types using Random Forest. June 2018. Big Earth Data 2(1):1-2

Xiong K, BR Adhikari, CA Stamatopoulos, Y Zhan, S Wu, Z Dong, B Di. Comparison of Different Machine Learning Methods for Debris Flow Susceptibility Mapping: A Case Study in the Sichuan Province, China, Remote Sensing 12 (2), 295 (2020)

Author Response

We would like to thank you very much for your comments and suggestions. We have tried our best to revise the manuscript, based on your suggestions and comments. Here are our responses to each of your comments and suggestions.

Reviewer 1:

Comment #1: The dimension of each different moving window symbols was based on the number of pixels, and the resolution of spatial data used in this study is 1 m.

Comment #2: Based on your suggestions, we have added more detailed discussions, your recommended references into the revised manuscript.

Comment #3: When we started data collection for this research, we also adopted the peak ground acceleration (PGA) data to analyse the effect of earthquake. However, the distribution of PGA was quite homogeneous in the study area during these 6 years. Therefore, the PGA was not included as a variable in advance in present study. Based on your suggestions, for our future long-term investigation, we will consider taking into account of strong earthquake events as a predictive variable into our analysis for landslide prediction.

Reviewer 2 Report

Review of the manuscript submitted to Remote Sensing entitled: ”Random forests for landslide prediction in Tsengwen River Watershed, Central Taiwan”.

The paper deals with landslide prediction using random forest algorithm. The paper appears interesting and it has a good structure and is written correctly, but my main criticism goes to the terminology. Based on my experience and literature, there are generally two main problems. First: existing landslide detection (based on remote sensing data) and landslide susceptibility mapping (assessing where landslide can occur in the future). Based on susceptibility mapping we can assess landslide hazards risk, hazards and make another analysis. The word “prediction” for me is referring to susceptibility of landslide mapping, and I would say that everything is ok in this manuscript. However, usually in landslide susceptibility mapping scientists are assessing various susceptibility zones (very low, low, moderate, high, very high), which is missing here.  I refer author to these two main review paper to analysis and specify what is the goal of their paper: landslide detection or susceptibility mapping? If the second, various susceptibility classes should be assessed. If first one, false positive and false negative should be presented on the map:

  • Reichenbach, P., Rossi, M., Malamud, B. D., Mihir, M., & Guzzetti, F. (2018). A review of statistically-based landslide susceptibility models. Earth-Science Reviews, 180, 60-91.
  • Guzzetti, F., Mondini, A. C., Cardinali, M., Fiorucci, F., Santangelo, M., & Chang, K. T. (2012). Landslide inventory maps: New tools for an old problem. Earth-Science Reviews, 112(1-2), 42-66.
  • Pawluszek-Filipiak, K., & Borkowski, A. (2020). On the Importance of Train–Test Split Ratio of Datasets in Automatic Landslide Detection by Supervised Classification. Remote Sensing, 12(18), 3054.--> (here false positive and false negative etc should be presented as on figure 9)

I really recommend this paper to be published in Remote Sensing, however in landslide literature there is really a need to correctly describe and name the studies in order to prevent any misunderstandings. At the moment in literature, there is also misunderstanding between the terms of susceptibility and hazards and it was discussed by Guzzetti et al., 2012. Therefore please made a careful review of the existing literature and use a proper terminology to you study.

In the present form, the paper needs of major revision before to be considered for publication in Remote Sensing.

Another observations are reported in the attached .pdf file.

Reviewer

Author Response

We would like to thank you very much for your comments and suggestions. We have tried our best to revise the manuscript, based on your suggestions and comments. Here are our responses to each of your comments and suggestions.

Reviewer 2:

Comment #1: Your comments and suggestions are very much appreciated. The objective of this study was to develop an approach for landslide prediction. To verify the accuracy of the model, we predicted landsides for 2015, and the results were thus verified by comparing the predicted landslide areas with the ground reference data collected in 2015. Based on your suggestions, we have carefully read and added your suggested articles, and accordingly improved our manuscript. The predicted results with the probabilistic values from 0.5 to 1 were into three classes to depict landslide susceptibility, including low susceptibility (0.5–0.7), moderate susceptibility (>0.7–0.9), and high susceptibility (>0.9–1) (Figure 4).

Comment #2:  Based on your suggestions, we have checked the recommended article, and added it into the revised paper. In this work, our objective was not to make comparisons of mapping accuracies between object-based and pixel-based algorithms for landslide prediction, rather additionally investigating the scale effects on different moving window sizes on the results. Your suggestions to include object-based analysis are very much appreciated, and we will consider this approach for our future investigations.

Comment #3: Regarding “the reference to the landslides are the seventh deadliest natural disaster”, we have checked the fact, and added references for this statement.

Comment #4: The full name of Tsengwen River Watershed (TRW) has been mentioned in the Abstract and Introduction sections. Thus, we only used the abbreviated TRW accordingly in the manuscript.

Comment #5: Based on your comments, we have revised Figure 1, by using the satellite image (SPOT-5 image) in 2015 to illustrate landslide areas.

Comment #6: Our comments regarding figure 2 and Line104 to106, we have revised Figure 2, as well as corrected the methods, and accuracy assessment section. In this study, the KNIME Analytics Platform was used to construct the RF prediction model. The data were divided into two groups of pixels, namely training pixels (30%), and evaluating pixels (70%) for accuracy assessment. For building the RF model, 50% of the group of training pixels were randomly extracted and used as training samples for model establishment, leaving 50% of the data for model validation. The pixel-by-pixel comparison between the predicted landslide areas and the ground reference data was also implemented using the group of evaluating pixels (i.e., 70% of the reference landslide map). To facilitate this comparison, the predicted landslide results were first categorized into two classes of landside areas, and non-landslide areas. The results of landslide prediction were then verified by comparing pixels from the group of pixels (used for accuracy assessment) with those synchronized from the RF prediction map. The error matrix using overall, producer’s, and user’s accuracies, and Kappa coefficient were calculated to measure the prediction accuracy.

Comments #7-8: Yes, in this study, we calculated predictive variables using ArcGIS’s toolsets and KNIME Analysis Platform program. We have improved the section of data preprocessing of variable creation and the qualify of related figures in the revised paper based on your suggestions.

Comments #9: We calculated the river density from DEM, using the hydrological tool in ArcGIS, following these 5 steps:

  1. Generating the flow direction from DEM data.
  2. Calculating the flow accumulation.
  3. Generating the flownet raster data.
  4. Transferring the flownet raster data to stream feature (polyline).
  5. Calculating the stream lines by using the Line Density tool in ArcGIS in different search radius.

Comments #10: Based on your suggestions “Line 118, it is not landslide modelling”, we modified it as “landslide prediction model”.

Comments #11: Your comments “Figure 3, color change ??? not visible”. We revised this figure, by correcting the class range and adjusting the character size of legend.

Comments #12: Line 221, This is really hard to believe, that you achieved 99% of accuracy.... are you sure that this are not parameters from cross validation rather that accuracy assessment based on ground truth? Based on reviewers’ comments, we have corrected the predicted map, and performed the accuracy assessment again. The results and discussions of the revised paper have been significantly improved.

Comments #13: Based on your suggestions, we have improved Figure 4.

Comments #14: Your comments: “Figure 5, legend of Misclassified, What it is? misclassified? false positives?”. We have improved Figure 5, as well as discussions related to commission and omission errors in the section of results and discussion. Specifically, we modified the presented results. The landslides susceptibility map was showed with the low, moderate and high probability. To validate the accuracy of landslides prediction, the high susceptibility zone was evaluated by the ground truth data in 2015.

Reviewer 3 Report

The authors have used random forest modeling to assess landslide hazards for a single watershed in Central Taiwan. Their model uses various hydrologic, geologic, and topographic data to classify pixels that are susceptible to landslides. The underlying data and landslide database were collected by others, so the main contribution of the current paper is the application of this method to a new area and an identification of the important factors for landslide susceptibility. I believe the paper could be of interest to researchers working on landslide hazard analysis, but I believe that more information is needed to judge the validity of the author’s approach. My comments are summarized below.

  1. What program or algorithm have the authors used for their random forest model? I don’t see a reference to any particular program or code. If the authors have implemented their own code, they should give the programming language and the reference for the algorithm they used.
  2. All of the data sources should be summarized in a table which includes the type of data, the source (e.g., aerial photography, satellite imagery, LiDAR, etc.), and the resolution. Some of this information is included in the text, but it needs to be summarized in a single location. This table could then replace much of the explanatory text in Section 4.2.
  3. The authors seem to have created a binary classification (landslide or no landslide), but I am much more familiar with probabilistic outputs from machine learning methods. These probabilistic outputs can be used to classify the likelihood of a landslide occurring, which is much more common for generating susceptibility maps. Why have the authors decided to use a binary classification?
  4. The most critical component of this work is the landslide inventory used for training the model, but it is not described in sufficient detail. Was the model only trained on landslides that occurred in 2015 (as stated in Figure 2)? What methodology was used to identify landslides? Was a minimum threshold set on landslide size? Where is this database documented and what “field survey data” (Line 85) was used to validate it?
  5. Lines 103-105: How were the pixels divided into these different groups? Were they randomly assigned or was the training done on a certain region of the watershed?
  6. Line 112: What is geologic line density? Why was geologic unit (or rock type) not included? Shale regions would likely have different susceptibility than sandstone.
  7. Line 122: What does contribution area mean? Is this the catchment area for the watershed?
  8. Line 134-135: How was this data for precipitation and evapotranspiration collected? Was this data only for the period in which the landslides occurred? This ends up being one of the most critical parameters so a thorough explanation is needed.
  9. Figure 3: This figure is very hard to see at this scale. I realize the authors are limited in terms of length, so could this be included as an appendix and the size increased?
  10. Lines 204-219: I believe some explanation of the various error measures is needed for audiences that are not familiar with machine learning? I know Receiver Operating Characteristic Curves are commonly used to judge predictive models. Have the authors considered using this method to evaluate their results?
  11. Figure 4: Does the green box show the larger figure in Figure 5? If so, this should be indicated. If not, the area shown in the top figure in Figure 5 should be highlighted. The caption should also explain that the letters refer to the areas in Figure 5.
  12. Figure 5: The green boxes seem to indicate that only a small area was misclassified, but there seems to be a large amount of yellow outside of the landslide footprints and a lot of unmarked pixels inside the landslide footprints. Aren’t all of these pixels misclassified? If so, is the authors’ reported error rate accurate?

Author Response

We would like to thank you very much for your comments and suggestions. We have tried our best to revise the manuscript, based on your suggestions and comments. Here are our responses to each of your comments and suggestions.

Reviewer 3:

Comments #1: In this study, we used the KNIME Analysis Platform program (section 4.1) to build the RF model, as well as perform the accuracy assessment.

Comments #2: Based on your suggestions to summary data sources and variables, we have improved this section by a table summarizing the data sources and predictive variables in Table 1 of the revised paper.

Comments #3: We have modified the predicted results of landslides as the landslides susceptibility map with the low, moderate and high probabilistic classes. To validate the accuracy of landslides prediction, we selected the class of high susceptibility (i.e., landslide and non-landslide) for accuracy assessment using the ground reference data. We have improved this section in the revised paper.

Comments #4: In present study, only the landslides occurred in 2015 were predicted. The ground truth data were constructed using high resolution satellite images during 2009 to 2015, where the areas greater than 0.1 hectares were verified by the Forestry Bureau, Council of Agriculture, and also field survey data from Central Geological Survey, Taiwan. We also improved this section in the revised paper.

Comments #5: Lines 103-105. The landslide pixels occurred in 2015 were randomly selected the 30% data for training and the other 70% data for accuracy assessment. We have revised the paragraphs related to methods of model establish and accuracy assessment in the revised manuscript.

Comments #6: Line 112: The geologic line density includes the locations of faults and folds as polylines, using the line density tool in ArcGIS to calculate the radius effect. In beginning of data collection, the lithology was considered as well. However, the distribution of lithology was quite homogeneous covered with sandstone and very little siltstone. Therefore, it is not included as a variable in this study.

Comments #7: Based on your suggestions, the term “contribution area” has been modified as “catchment area”.

Comments #8: Lines 134-135. The precipitation and evapotranspiration data were collected from the 14 weather stations across the study area. In this study, 2011-2014 data were collected and calculated the annual average precipitation data to estimate the annual groundwater recharge.

Comments #9: Thanks for your comments and suggestion regard “figure is very hard to see at this scale”, we have revised and improved the quality of Figure 3, by correcting the class range and adjust the character size of legend.

Comments #10: Lines 204-219. As your suggestion, the area under the curve (AUC) of receiver operating characteristic (ROC) has been added into the sections of accuracy assessment, and results and discussion.

Comments #11: As you commented on Figure 4, we have improved this figure. The green box in Figure 4 was the high density landslides area and also showed in Figure 5.

Comments #12: Your comments related to the green box, we have improved this figure, and also improved the section of results and discussion, by modifying the landslides susceptibility map with the low, moderate and high probability. To validate the accuracy of landslides prediction, the high susceptibility zone was evaluated by the ground truth data. The susceptibility map of (A), (B) and (C) enlarged scale of landslides was compared with the landslides ground truth in 2015.

Round 2

Reviewer 3 Report

The authors have adequately responded to my previous comments and I believe the paper has been improved. My only remaining comments would be to add units to the radii (not radiuses) listed in Table 1 and to remove the large blank space on Lines 165-166.